# Walk Locomotion Kinematic Changes in a Model of Penetrating Hippocampal Injury in Male/Female Mice and Rats

**DOI:** 10.3390/brainsci13111545

**Published:** 2023-11-02

**Authors:** Jonatan Alpirez, Lilia Carolina Leon-Moreno, Irene Guadalupe Aguilar-García, Rolando Castañeda-Arellano, Judith Marcela Dueñas-Jiménez, Cesar Rodolfo Asencio-Piña, Sergio Horacio Dueñas-Jiménez

**Affiliations:** 1Departamento de Neurociencias, Centro Universitario de Ciencias de la Salud, Universidad de Guadalajara, Guadalajara 44340, Mexico; jonatan.alpirez@gmail.com (J.A.); lica.leonm@hotmail.com (L.C.L.-M.); irene.agarcia@academicos.udg.mx (I.G.A.-G.); 2Centro de Investigación Multidisciplinario en Salud, Centro Universitario de Tonalá, Universidad de Guadalajara, Tonalá 45425, Mexico; rolando.castaneda@academicos.udg.mx; 3Departamento de Fisiología, Centro Universitario de Ciencias de la Salud, Universidad de Guadalajara, Guadalajara 44340, Mexico; judith.duenas@academicos.udg.mx; 4Departamento de Electronica, Centro Universitario de Ciencias Exactas e Ingenierias, Universidad de Guadalajara, Guadalajara 44430, Mexico; rodolfo.ascencio@academicos.udg.mx

**Keywords:** locomotion kinematics, hippocampal injury, mice, rats, sex differences

## Abstract

Traumatic brain injury has been the leading cause of mortality and morbidity in human beings. One of the most susceptible structures to this damage is the hippocampus due to cellular and synaptic loss and impaired hippocampal connectivity to the brain, brain stem, and spinal cord. Thus, hippocampal damage in rodents using a stereotaxic device could be an adequate method to study a precise lesion from CA1 to the dentate gyrus structures. We studied male and female rats and mice, analyzing hindlimb locomotion kinematics changes to compare the locomotion kinematics using the same methodology in rodents. We measure (1) the vertical hindlimb metatarsus, ankle, and knee joint vertical displacements (VD) and (2) the factor of dissimilarity (DF). The VD in intact rats in metatarsus, ankle, and knee joints differs from that in intact mice in similar joints. In rats, the vertical displacement through the step cycle changed in the left and right metatarsus, ankle, and knee joints compared to the intact group versus the lesioned group. More subtle changes were also observed in mice. DF demonstrates contrasting results when studying locomotion kinematics of mice or rats and sex-dependent differences. Thus, a precise lesion in a rodent’s hippocampal structure discloses some hindlimb locomotion changes related to species and sex. Thus, we only have a qualitative comparison between murine species. In order to make a comparison with other species, we should standardize the model.

## 1. Introduction

The hippocampus is a brain structure extensively associated with mental processes involved in memory and emotional regulation, space navigation, and cognitive processes [1,2]. It is associated with the sensory-motor system in murine models. The hippocampus is related to gait locomotion and control [3,4,5,6] and locomotion speed changes [7,8,9,10]. Some studies reported locomotion kinematics disruption after hippocampal injury in both rats [5] and mice, and these effects were absent with the administration of treatments for hippocampal neuroprotection as after intranasal administration of endometrial mesenchymal stem cell-derived extracellular vesicles charged with progenitor cells [11,12]. Recent studies use intrahippocampal injection of stem cells of GABAergic interneurons to avoid seizures in mice [13], but should be made in rats.

The results obtained from previous studies could not be compared due to using different methodologies and animals. Divergences in the anatomy and physiology of mice and rats have been previously described [14], and the movement in rats is sex-dependent [15,16]. It can also result from an evolutionary brain development [17], leading to different results even using the same model. Stride length, mechanical behaviors, muscle strength and contraction, and movement differences occurred between male and female murine due to sex hormones and sex differences in CNS. Anatomy and behavior, or muscle architecture differences, appear in complex motor behaviors [16,18,19,20,21]. Also, sex-dependent behavior and brain damage differences have been extensively described after brain injury and neurodegenerative diseases [11,22,23,24,25,26]. A sex difference also occurs in the stroke vulnerability in C57BL/6 mice [27]. However, a quantitative standard method has not determined subtle locomotor differences in each step between males and females and rats or mice [28].

Consequently, we use a detailed methodology to study the effect of the hippocampus using a penetrating hippocampal injury model to understand further how the hippocampus is related to changes in vertical hindlimb joint displacement during locomotion. In future experiments, we will use medication or a prosthesis to correct the feedback system of the vertical displacement. We also analyzed the dissimilarity factor to evaluate a kinematic parameter between rats and mice to analyze the simple elements of complex locomotor behavior. 

## 2. Materials and Methods

### 2.1. Animals and Experimental Study Groups 

All animal experiments were carried out under the UK Animals (Scientific Procedures) Act, 1986 and associated guidelines, EU Directive 2010/63/EU for animal experiments, the Mexican Regulation of Animal Care and Maintenance (NOM-062-ZOO-1999, 2001), and the internal University of Guadalajara regulations. Animal use was approved by the Institutional Bioethical Committee for Institutional Animal Care and Use (IACUC), with an approved code for use in laboratory animals (23–42, v.2). We include adult Wistar rats (180–250 g, 10–12 weeks) and C57BL6 mice (20–25 g, 6–8 weeks). We put the animals in groups in plastic cages with a 12 h light–dark cycle at room temperature (22 °C ± 2 °C) and constant humidity water and food were available ad *libitum* For the analyses, we divided the animals into eight groups, five animals each: control rats: males (CMR) and females (CFR), control mice: males (CMM) and females (CFM) Later, we used the same rats and mice Lesioned rats: males (LMR) and females (LFR), and lesioned mice: males (LMM) and females (LFM). We used the pre-injury data of each experimental group as a control.

### 2.2. Brain Injury Model

We made a penetrating lesion in the hippocampus to obtain our injury model. Animals were anesthetized with isoflurane gas at 5% for induction and 2–3% for performed surgeries under aseptic conditions. After verifying the absence of pain reflexes, the experimental subject was placed in a stereotaxic frame, and a head incision over the middle line was made. We performed the brain injury model as described before for rats [5] and mice [9].

Briefly, a cleft was drilled into the left parietal bone to expose the meninges. A 0.5 mm diameter sterile steel cannula was placed at the coordinates from Bregma in mice (ML: −1 mm, AP: −2.5 mm) and rats (ML: −2 mm, AP: −5 mm), according to Paxinos and Franklin and Paxinos and Watson brain atlas coordinates, respectively. The cannula penetrated a depth of 2 mm (mice) and 4 mm (rats). Then, we displaced it rostrally 1 mm (mice) and 2 mm (rats) and pulled it out. We applied pressure to the injury to stop the bleeding with gauze; after the procedure, we sutured the scalp incision with 4–0 nylon sutures using 2% lidocaine in the injury. After the surgery, animals were placed in a heat blanket for 30 min until they woke up and moved freely. Then, we administered antibiotics (Enroxil 2 mg/kg, Sinosiain, Zapopan, Mexico) and analgesics (meloxicam 2 mg/kg, Pisa Laboratory, Guadalajra, Mexico) for three days.

### 2.3. Tunnel Walk Recordings 

Kinematic records of the animals’ unrestrained gait locomotion were made on day 0 and 7 days after the injury. We identified the videos on day 0 as the control groups (before the injury) and those on day 7 as the lesioned groups. Video recordings of the animals while walking on a transparent Plexiglas tunnel (tunnel walk) were taken. We used two synchronized cameras that recorded left and right hindlimbs simultaneously at 240 fps with a resolution of 1280 × 720 pixels. Post-processing was applied to the resulting videos to remove spherical distortion due to the lenses by estimating a homographic matrix using four points on the image [25]. We selected the animal steps of interest using the instants corresponding to the beginning and end of the step. To obtain the displacement curves of each point of interest (knee, ankle, and metatarsus), we manually annotate them (Figure 1A) on each video frame for each step using custom-made software. We analyzed the values in software developed in our laboratory to assess the kinematic analyses; each of the steps of the animal captured on the video was analyzed separately.

### 2.4. Dissimilarity Factor and Vertical Displacement Analysis 

To assess locomotion kinematic pattern analysis from each point, we generated displacement curves on the horizontal and vertical axes concerning time for each point in the left and right hind limbs during several steps. All curves were normalized according to the stride using a value range from 1 to 100, employing a spline-based interpolation. 

We compared their displacement curves to determine dissimilarity factor (DF) changes between animal groups. We calculated the difference between them using the Euclidean distance between each of the points of the normalized curve on the horizontal (X) and vertical (Y) axes as described before [9] (Appendix A).
$$DF_{<a,b>}=\frac{1}{200}\sqrt{\sum_{i=1}^{100}{\left(x_{a}(i)-x_{b}(i)\right)}^{2}+\sum_{i=1}^{100}{\left(y_{a}(i)-y_{b}(i)\right)}^{2}}$$
where *DF*_<*a*,*b*>_ is the squared error between every point of the normalized curves, defined as difference factor (*DF*); “*x_a_*(*i*) − *x_b_*(*i*)” is the difference between the coordinates in *x*, and “*y_a_*(*i*) − *y_b_*(*i*)” in *y* of every point in the graph, when comparing two steps (a and b); and “*i*” is the percent in the step cycle. 

We compared all the curves of every animal in the control and the lesioned groups seven days post-injury. Every step curve of an animal (a) was compared to the curves of a different control animal b, sequentially with each displacement curve of all animals in the control group. Thus, we calculated differences in the displacement curves between every animal and compared experimental versus control groups. We graphed all values and statistical hypothesis tests to evaluate group distribution differences.

Moreover, we averaged the vector components at each point of the normalized displacement curves for each group. We then determined significant differences between groups using a student’s *t*-test (a = 0.05) for each step phase point for both vertical (Y) and horizontal (X) components. We compared this pattern comparison analysis by using a locally designed Matlab script.

### 2.5. Statistical Analysis 

The Kolmogorov–Smirnov test was used to determine data normality. All results are expressed as means ± SEM. For DF analysis, we used a Kruskal–Wallis test with Dunn post hoc, and for vertical displacement analysis, we used an unpaired one-tailed student *t*-test. A value of *p* < 0.05 was considered statistically significant. We made the Statistical analysis using the Prism 9.0 software (GraphPad, Boston, MA, USA).

### 2.6. Histological Analysis

Once the rats’ kinematics were recorded, the rats were anesthetized intraperitoneally with pentobarbital at 160 mg/kg of body weight. They were sacrificed by intracardiac perfusion with 0.9% saline solution followed by 4% paraformaldehyde in phosphate buffer 0.01 M (pH 7.4) Brains were removed and placed in the same fixed solution at four °C We made brain coronal vibratome sections of twenty micrometers (Thermo Scientific, HM650V, Waltham, MA, USA) To identify the hippocampus region affected by the penetrating injury, we utilized cresyl violet staining to visualize this specific area. 

We immerse samples in absolute ethanol for 10 min (Twice), then 10 min in 96° ethanol, followed by 10 min in 80° ethanol, then 10 min in 50° ethanol, 5 min in distilled H_2_O, and 15 min in 0.1% cresyl violet solution After staining, we wash them in distilled water and place them in 96° ethanol for several minutes Later, it was submerged in absolute ethanol and xylol Finally, a mounting medium was added and observed under the light microscope.

In mice, at 7 DPI, mice were anesthetized with xylazine/ketamine [intraperitoneal (IP), 80 mg/kg] and intracardially perfused with phosphate-buffered saline (PBS) and fixated with 4% paraformaldehyde in PBS. The brain was immediately removed and postfixed for 48 h at four °C. The paraffin-embedded area of interest (−1.5 to −2.5 mm from Bregma) was then sliced tissue into 5 mm thick sections. Approximately 40 slices were obtained and mounted into positively charged slides. We used hematoxylin–eosin staining to assess the extension of the penetrating brain injury. A representative sample of nine brain sections was rehydrated in alcohol solutions and then submerged in a 5% hematoxylin solution for 3 min, followed by a 2% eosin solution for 30 s. Slides were finally washed, dehydrated, and sealed with Entellan resin. We observed the brain tissue slices using a light microscope (Figure 1B).

## 3. Results

In all lesioned rats, after a histological examination of the hippocampus, we found the lesion is going from the CA1 region to the dentate gyrus both in rats and mice Figure 1A. We also observed damage and necrotic area as indicated by sorrow heads as indicated in Figure 1B. 

### 3.1. Vertical Displacement in Hindlimbs Metatarsus, Ankle, and Knee in Rats and Mice

The metatarsus VD is different in intact rats than in intact mice; in rats, it is executed in one direction (Figure 2A–D), while in mice, it occurs in two directions (Figure 2F–H). 

In male lesioned rats, the left metatarsus VD changed 35% of the step cycle compared to intact rats (Figure 2A). In the right metatarsus, the change was 9% (Figure 2B). Female rats did not show vertical displacement changes on the left metatarsus versus intact rats (Figure 2C). However, the change versus control in the right metatarsus was 8% of the step cycle (Figure 2D). 

In mice, the vertical displacement analysis reveals changes in the left side of LMM compared to CMM in 8% of the step cycle (Figure 2E). On the right side, the changes were in 6% of the step cycle (Figure 2F). Comparing CFM versus LFM, the vertical displacement of the left side changed in 4% of the step cycle (Figure 2G). On the right side, we observed no changes (Figure 2H). In Figure 2A–H, all bins with symbols above zero indicate a significant statistical difference. Thus, a feedback system could correct each bin difference concerning the step cycle curve in non-lesioned animals. 

### 3.2. The Ankle VD in Intact Rats Differs from That in Intact Mice

In mice, it is a pendular displacement (Figure 3E–H), while in rats, it is executed only in one direction (Figure 3A–D). In rats, the vertical displacement of the left ankle comparing CMR versus LMR was different. The step cycle changed by 63% (Figure 3A). It also changed 62% of the step cycle on the right side (Figure 3B). In the female rat group, the vertical displacement of the ankle changes by 10% of the step cycle on the left side (Figure 3C) and 22% on the right side (Figure 3D). 

In mice, the vertical displacement of the ankle showed changes between LMM and CMM in 18% of the step cycle on the left side (Figure 3E) and 6% on the right side (Figure 3F), While the changes between LFM and CFM were 16% and 2% of the step cycle in the left (Figure 3G) and right ankle (Figure 3H), respectively.

### 3.3. The Knee VD Differs between Intact Rats versus Intact Mice

In rats, it occurred in a pendular-like movement (Figure 4A–D), which did not occur in knee mice displacement, which is unidirectional (Figure 4E–H). In rats, the vertical displacement of the knee reveals changes in both sides of males and females. The step cycle LMR changes by 9% (Figure 4A). The CMR on the left hindlimb and 4% on the right hindlimb (Figure 4B). In females, the step cycle of the LFR differs by 8% from the CFR (Figure 4C) and 42% on the right knee (Figure 4D). In mice, the vertical displacement of the knee revealed no changes on the left side of CMM vs. LMM (Figure 4E), but on the right side, 4% of the step cycle was different (Figure 4F). The vertical displacement analyses of LFM show changes in the step cycle of 22% on the left side (Figure 4G) and 6% on the right side (Figure 4H).

### 3.4. DF in Rats and Mice’s Metatarsus, Ankle, and Knee Joints

The left metatarsus DF values in rats were not statistically significant in CMR versus LMR groups (Figure 5A). In contrast, CFR and LFR’s DF values have statistically significant differences (*p* = 0.039, Figure 5A). Attending to sex, CMR versus CFR and LMR versus LFR have a DF statistically significant difference (*p* = 0.0062 and *p* = 0.0378, respectively. In the right metatarsus, there was no statistical difference between CMR versus LMR (Figure 5B). The DF value CMR vs. CFR has statistically significant differences (*p* = 0.0003, Figure 5B). Statistical significant differences also occurred among Thus, in male rats, the hippocampus lesion did not play an important role when the hindlimbs touched the ground in each stride. In mice, the DF value in the left metatarsus of lesioned males made no significant differences compared to controls (Figure 5C). In the right metatarsus, we found statistically significant differences between CMM and LMM (*p* < 0.000001). In the analyses of sex, we found, on the left metatarsus, that the DF value statistically significantly differs between CMM and CFM (*p* = 0.000619) and between LMM and LFM (*p* = 0.000194). We also found statistically significant differences between CMM and CFM (*p* = 0.000013) on the right side. Thus, the mouse hippocampus lesion affects the stride configuration only in the right hindlimb.

In rats, in the left ankle, the DF, between CFR and LFR, did not have statistical differences. In contrast, the DF value of CMR versus LMR was significantly different (*p* = 006342, Figure 6A). In the right ankle, we observed a DF statistically significant between CFR versus LFR (*p* = 0.005869). We also observed a DF statistically significant value when we evaluated CMR versus CFR groups (left ankle, *p* = 0.005913; right ankle, *p* =0.005367). In LMR versus LFR groups (left ankle, *p* = 0.000012; right ankle, *p* = 0.027108).

In mice, the DF values in the left ankle were different in lesioned mice compared to the control group in both males (*p* = 0.039045) and females (*p* = 0.000001) (Figure 6C). No differences occur when comparing control vs. lesioned groups on the right side (Figure 6D). When we compared male and female groups to each other, we observed differences among controls in the left ankle (*p* = 0.000001) and in both controls (*p* = 0.000001) and lesioned (*p* = 0.000040) in the right ankle. 

The knee DF analysis in the rat shows that no significant changes on the left side in either male or female groups (Figure 7A), but on the right side, the changes were statistically significant were between male vs. female control groups (*p* = 0.000023) Moreover, in rat female groups, the lesioned group shows significant changes compared with the control group (*p* = 0.000029) (Figure 7B).

Otherwise, in mice groups, the changes occurred on both sides; on the left side, the control female group was different from the control male group (CMM) (*p* = 0.000001) and the lesioned female group (LFM) (*p* < 0.000001) (Figure 7C). Additionally, statistical changes were found between lesioned males and lesioned females (*p* = 0.000710). On the right side, the changes occurred when females vs. males were compared in the control groups (*p* = 0.000029) and the lesioned groups (*p* = 0.000093) (Figure 7D).

## 4. Discussion

Damage in the hippocampus has been studied using different methods to produce a hippocampal lesion, resulting in alterations in posture, navigation, and locomotion. 

These different methods as early postnatal hypoxia [26], Rx in the dentate gyrus in adult rats [29], NMDA lesions [6], colchicine injected in the dentate gyrus [2], Rx directed to the hippocampus in neonatal rats [30]. Other studies use drugs such as fentanyl in male and female rats during complex motor tasks [21] or analyze the kinematics of treadmill locomotion in mice raised in hypergravity. Kinematic locomotion differences between male and female rats were studied in complex movements such as evading dodging movements [16,24]. It is also studied in a transgenic Alzheimer’s model, analyzing gender differences comparing transgenic rats versus control rats. Recently, Shepphard studied the mouse stride in an open field using deep learning [30]. Our study used female and male rats and mice walking in a transparent glass tunnel. 

We used a stereotaxic lesion going from the C1-throw dentate gyrus. We analyzed the VD and DF as stride components during locomotion. Therefore, a comparison with previous models is still being determined.

The VD of the left metatarsus in female rats and knee female mice was similar in control versus lesioned animals. The other joints have variable changes in different epochs of the step cycle, either in rats or mice. This lesion produces a different change in hindlimb movements attributed to the lesion size or subtle changes in hippocampus connectivity. It is also possible that the lesion produces changes in the connectivity between the hippocampus and the burst generators working independently for each hindlimb joint [31,32]. It will be interesting to study the hippocampus asymmetrical persistent theta rhythm related to locomotion and whether it is also manifested in spinal cord central pattern generators (CPG) in intact and lesioned male and female rats and mice. It analyzes its influences on CPG in hindlimb movements [10] and locomotion speed [4].

The DF values were highly variable, considering that it was studied in these groups. Thus, the several components configuring a stride are related to the neural circuitry employed in the diverse types of locomotion. However, it is interesting that the hippocampus lesion did not alter the DF values in male rats in the left and right metatarsus. This finding is at odds with previous work where estrogens protect female rats more [33]. 

Interestingly, the statistically significant differences among the different groups are similar in the metatarsus of left rats and mice hindlimb. In rats and mice, possible damage produced by the lesion in a similar hippocampal region could explain whether a similar pathway sends information to the metatarsus CPG. 

In rats, the DF values were statistically significant, comparing intact males versus intact females in the left and right metatarsus, the left and right ankle, and the right knee. In mice, we also observed DF values of metatarsus, ankle, and knee joints in intact male mice versus intact females with statistically significant differences. Subtle differences in animal size or undetectable skin slippage moving the ink mark could contribute to these changes. Different muscle forces in slow muscles in males versus females, particularly for knee and ankle joint movements, are an additional factor in explaining these changes [20].

The DF has a statistical difference in the ankle joint between rats and mice and between left and right hindlimbs. This statistically significant variation also occurs between intact or lesioned rats or mice. There were no statistically significant changes in the lesioned right or left ankle joints in mice.

The rat’s left knee had no statistical significance among the different groups. Differences occurred in this joint in mice compared intact male versus female mice and between control versus lesioned female mice. In the right knee, they appear between intact male versus female rats and intact male versus female mice. We observed differences between lesioned male and female mice in the left and right knee. The high variability of changes does not seem to be related to the model species or sex but to the damage of different hippocampus to spinal cord pathways, a different process of dentate gyrus progenitor cell proliferation, or a recovery in afferent inputs to the lesioned C1-dentate gyrus [34]. 

## 5. Conclusions

The accuracy of VD hindlimbs displacement is a precise variable to obtain information on each segment of the joint movements of the step cycle in order to develop an external device to train animals or patients with hippocampal lesions. The device could be a mechanical control system or an electrical muscle stimulator programmed to deliver adequate stimulation to reverse the changes. The DF could be used to quantify the stride in clinical studies and probe some pharmacological or other therapies.

## Figures and Tables

**Figure 1 brainsci-13-01545-f001:**
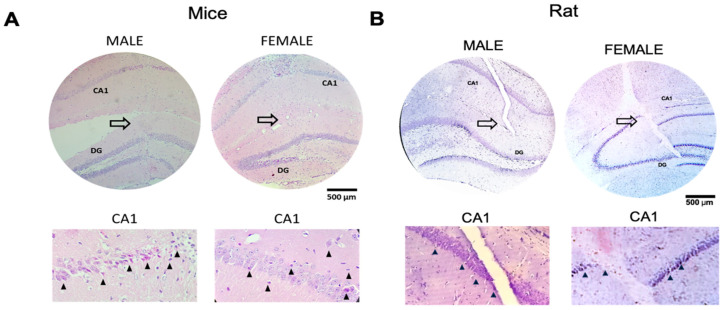
Histology of the hippocampus showing the lesion area (big white arrow), as well as its approach from the CA1 region and dentate gyrus damaged in mice and rats, both female and males (**A**,**B**). The arrowhead shows, in the CA1 magnification, the necrotic cells.

**Figure 2 brainsci-13-01545-f002:**
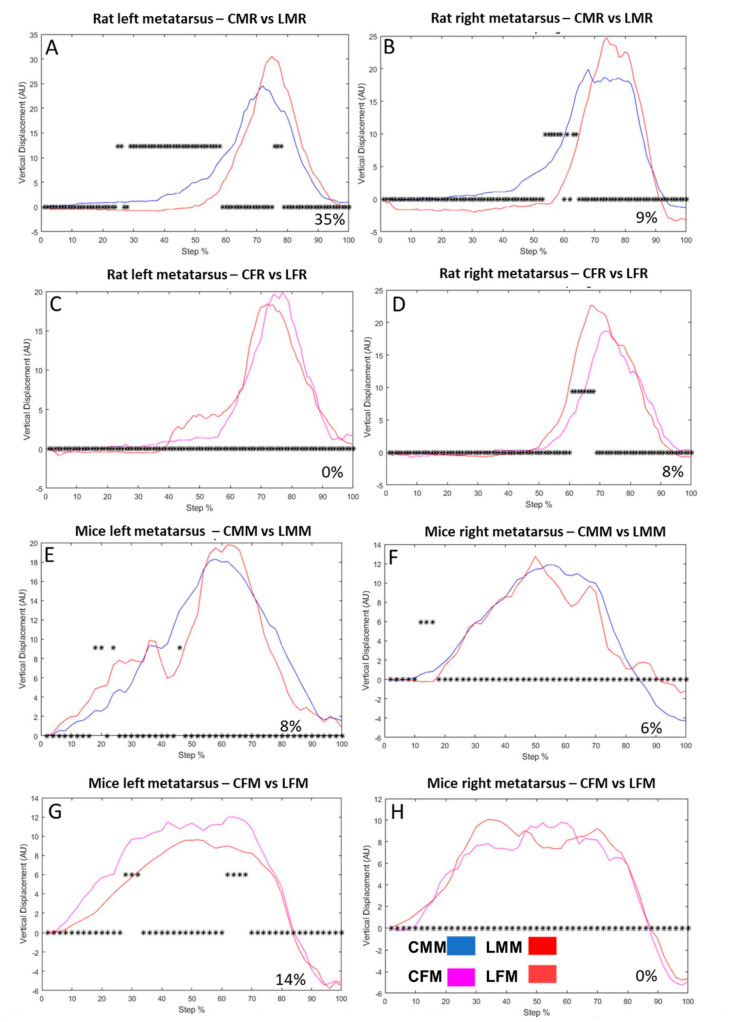
Male rat metatarsus vertical displacement (VD) in left (**A**) and right (**B**) hindlimbs Ordinate: vertical displacement in arbitrary units (AU), abscise: the step cycle was divided in 100 bins (cycle percentage), asterisks above zero indicate bins with a significant statistical difference (**C**,**D**): illustrate the vertical displacement in control and lesioned groups of female rats on the left and right sides, respectively (**E**,**F**): Shows mice male vertical displacement of CMM and LMM groups on the left and right sides, respectively (**G**,**H**): VD of CFM and LFM groups of the left and right sides, respectively.

**Figure 3 brainsci-13-01545-f003:**
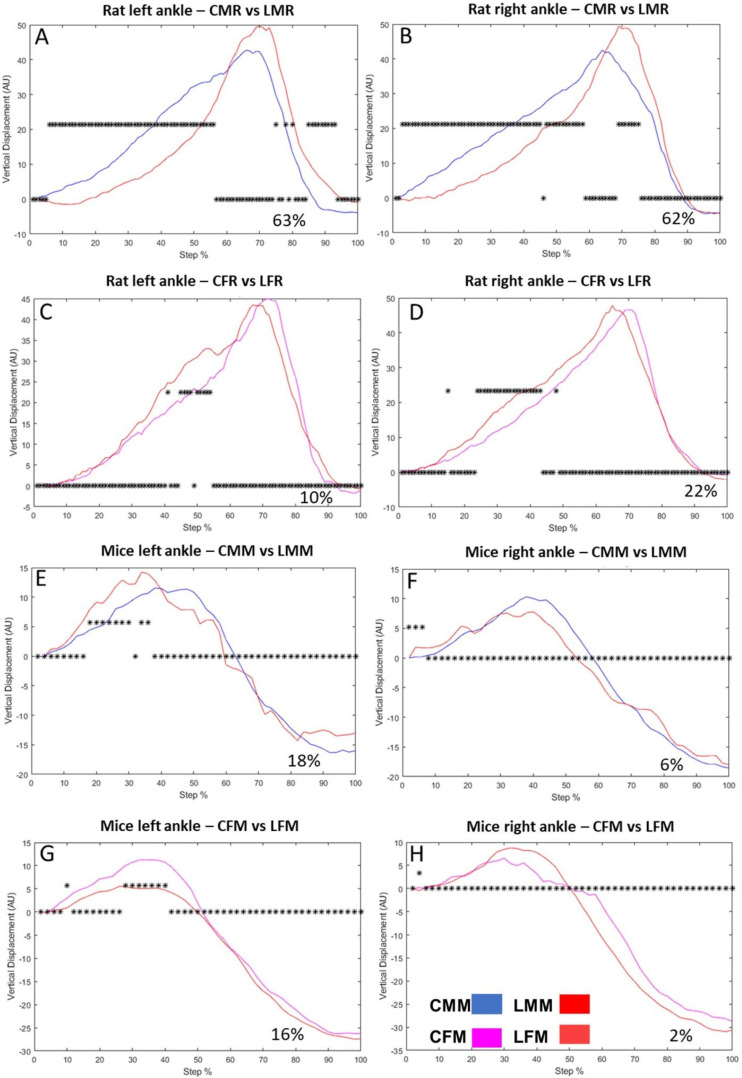
Male rat ankle vertical displacement (VD) in the left (**A**) and right (**B**) hindlimbs Ordinate: vertical displacement in arbitrary units (AU), abscise: the step cycle was divided in 100 bins (cycle percentage), asterisks above zero indicate bins with a significant statistical difference (**C**,**D**): illustrate the vertical displacement in control and lesioned groups of female rats on the left and right sides, respectively (**E**,**F**): Shows mice male vertical displacement of CMM and LMM groups on the left and right sides, respectively (**G**,**H**): VD of CFM and LFM groups of the left and right sides, respectively.

**Figure 4 brainsci-13-01545-f004:**
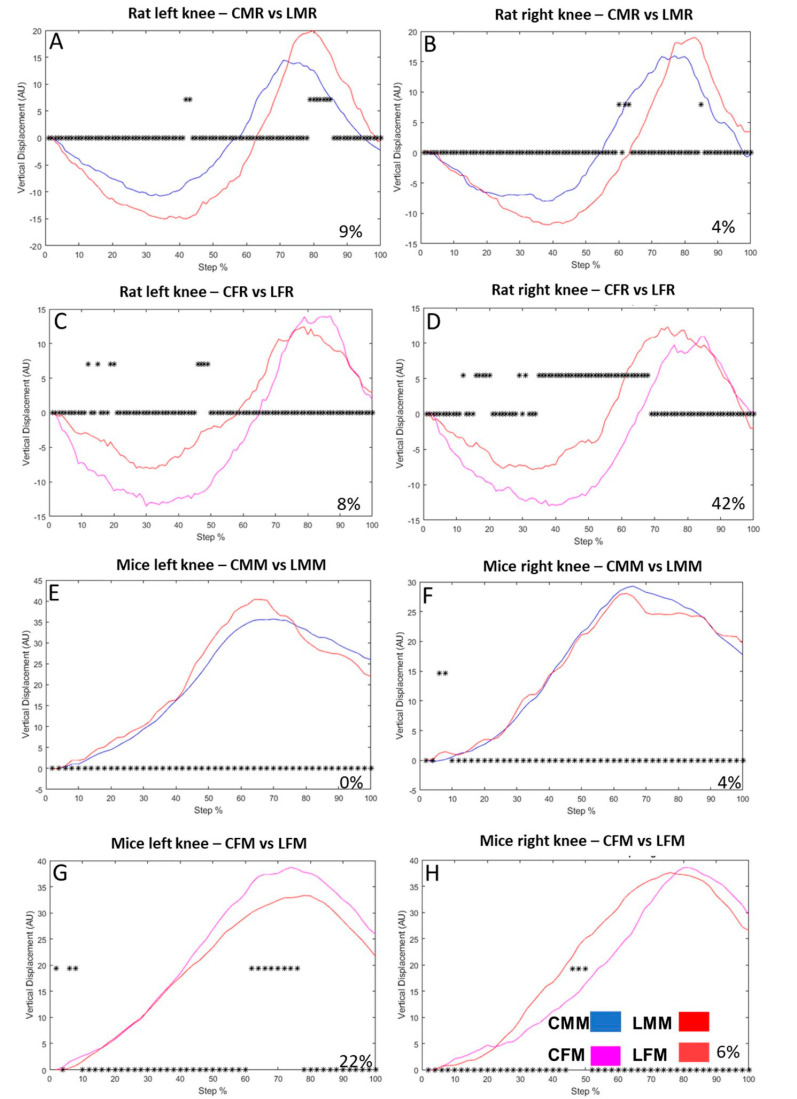
Figure 3 Male rat knee vertical displacement (VD) in the left (**A**) and right (**B**) hindlimbs Ordinate: vertical displacement in arbitrary units (AU), abscise: the step cycle was divided in 100 bins (cycle percentage), asterisk above zero indicates bins with a significant statistical difference (**C**,**D**): illustrate the vertical displacement in control and lesioned groups of female rats on the left and right sides, respectively (**E**,**F**): Shows mice male vertical displacement of CMM and LMM groups on the left and right sides, respectively (**G**,**H**): VD of CFM and LFR groups of the left and right sides, respectively. Appendix A is added to illustrate the VD in metatarsus, ankle, and knee joints, both in rats and mice.

**Figure 5 brainsci-13-01545-f005:**
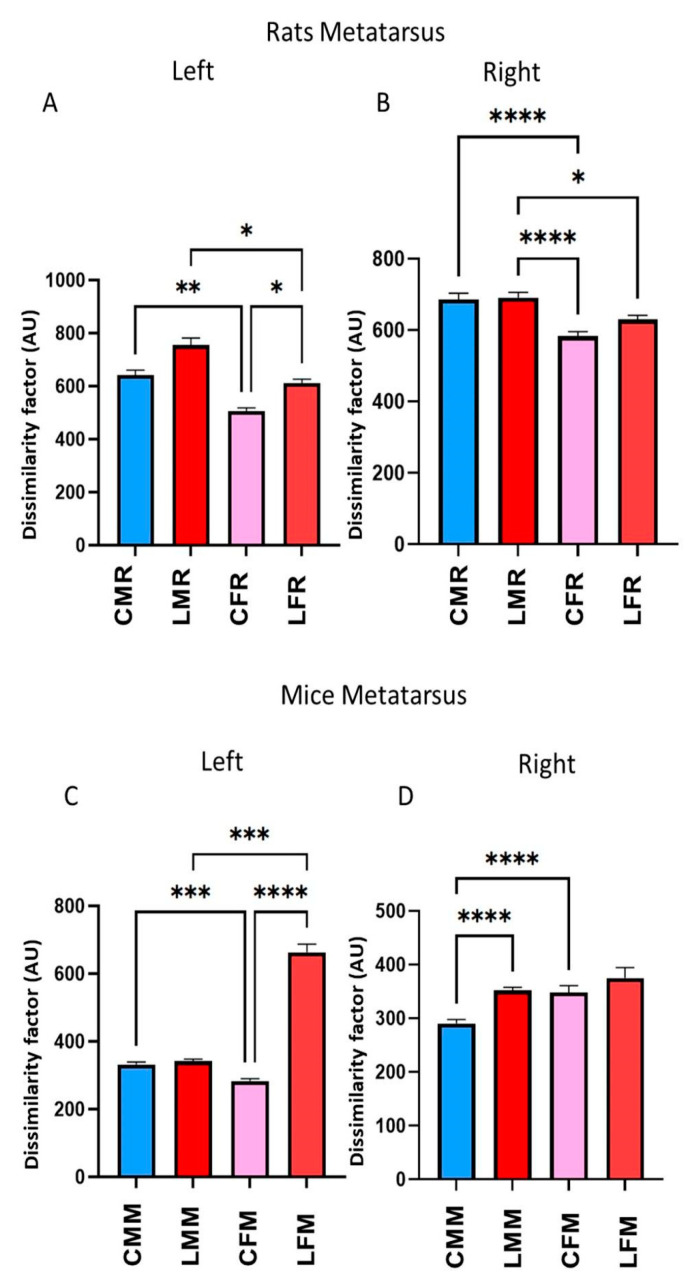
(**A**,**B**), rats dissimilarity factor (DF) in the left and right metatarsus joint. The bars in the graphic illustrate CMR, LMR, CFR, and LFR rat groups. The metatarsus vertical (Y-axis) displacement was averaged. Compared between control and lesioned rats, the Corresponding bins of each rat were averaged (n = 5). Significant statistical differences (*p* ≤ 0.05) were obtained using a T-student test (**C**,**D**), mice dissimilarity factor (DF) in the left and right metatarsus. The bars in the graphic illustrate CMM, LMM, CFM, and LFM rat groups. They show Mean ± EE values. We calculate statistical differences between groups using the ANOVA test *p* ≤ 0.05 (*), *p* ≤ 0.01 (**), *p* ≤ 0.001 (***), *p* ≤ 0.0001 (****).

**Figure 6 brainsci-13-01545-f006:**
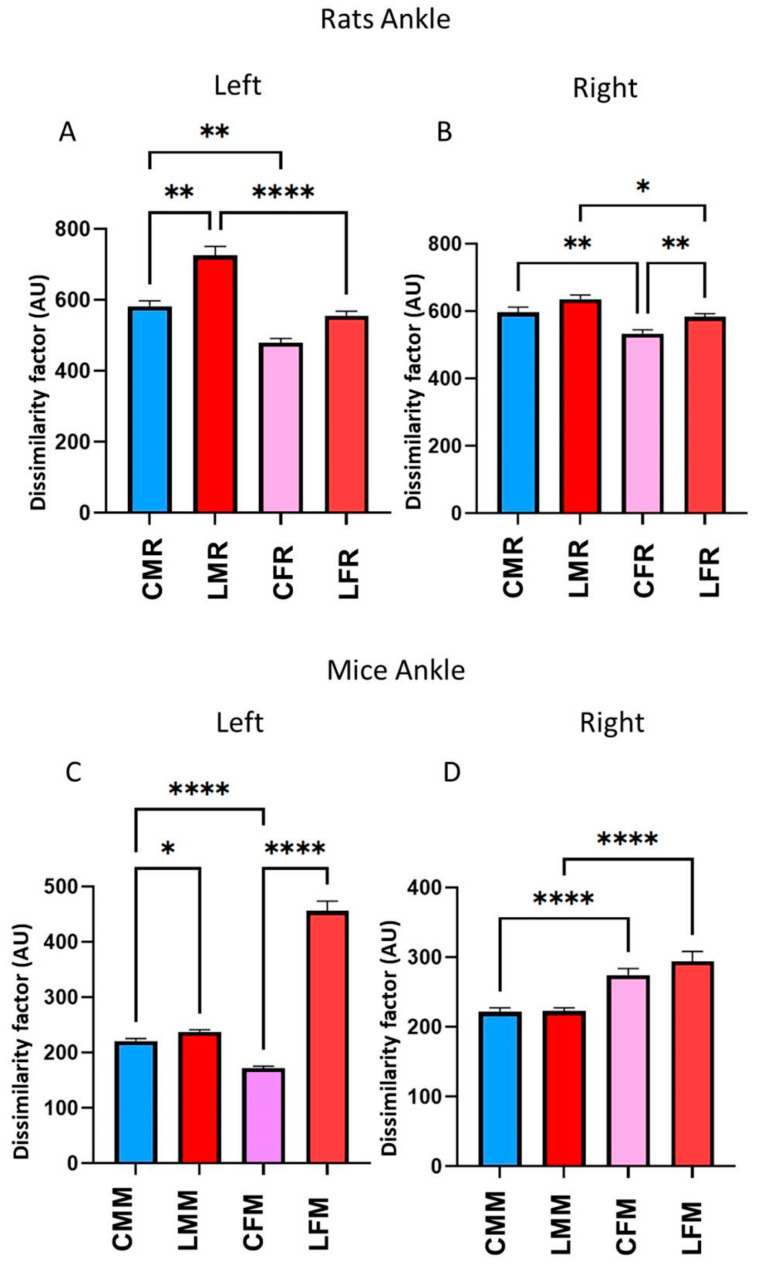
(**A**,**B**) rats dissimilarity factor (DF) in the left and right ankle joints. The bars in the graphic illustrate CMR, LMR, CFR, and LFR rat groups. The metatarsus vertical (Y-axis) displacement was averaged and compared between control and lesioned rats. Corresponding bins of each rat were averaged (n = 5), and significant statistical differences (*p* ≤ 0.05) were obtained using a T-student test (**C**,**D**), mice dissimilarity factor (DF) in the left and right ankle. The bars in the graphic illustrate CMM, LMM, CFM, and LFM rat groups. They show Mean ± EE values. The statistical differences between groups were calculated using the ANOVA test *p* ≤ 0.05 (*), *p* ≤ 0.01 (**), *p* ≤ 0.0001 (****).

**Figure 7 brainsci-13-01545-f007:**
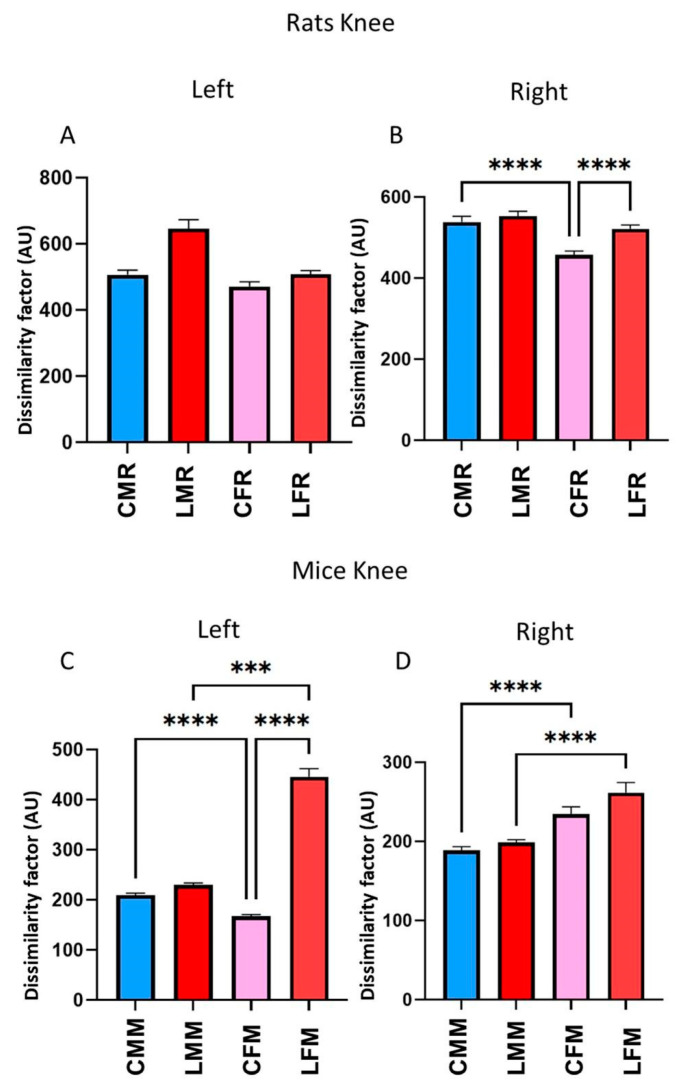
(**A**,**B**) rats dissimilarity factor (DF) in the left and right knee joints. The bars in the graphic illustrate CMR, LMR, CFR, and LFR rat groups. (C,D) mice dissimilarity factor (DF) in the left and right metatarsus by knee. They show Mean ± EE values. The statistical differences between groups were calculated using the ANOVA test *p* ≤ 0.001 (***), *p* ≤ 0.0001 (****).

## Data Availability

The raw data supporting the conclusions of this article will be made available by the authors by request without reservations.

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
