# Peer review of "Walk Locomotion Kinematic Changes in a Model of Penetrating Hippocampal Injury in Male/Female Mice and Rats"

_brainsci, 2023, doi:10.3390/brainsci13111545_

Round 1
Reviewer 1 Report
Comments and Suggestions for Authors
The paper titled " Walk Locomotion Kinematic Changes in a Model of Penetrating Hippocampal Injury in Male/Female Mice and Rats" by Alpirez et al. examines alterations in the locomotion patterns of mice and rats following a penetrating injury to the hippocampus. The authors employ video recording and hindlimb locomotion analyses to assess modifications in various stages of movement subsequent to brain lesions. This manuscript encompasses two rodent species, mice and rats, and scrutinizes distinctions between them, as well as disparities between males and females within each species. Their findings reveal notable variances between species and sexes. The authors discuss how their results indicate that vertical displacement in hindlimbs and the factor of dissimilarity are vital metrics for evaluating hippocampal injury and responses to treatment.
However, the paper exhibits several inadequacies that require rectification prior to consideration:
1. The organization of results is challenging to follow due to the numerous variables. The authors present data from hindlimbs on both sides of the lesion, for both males and females, and for both mice and rats. The presentation of results is intricate, and it is difficult to discern the rationale for some of the comparisons made. The manuscript's impact could be significantly enhanced by reorganizing the data presentation.
2. In Figure 1, it is unclear what is being demonstrated in panel B. Arrows and arrowheads are present in the histological images, but their significance is not explained. Furthermore, the meaning of panels C and D is unclear, and it is unclear whether the differences in the plots are statistically significant.
3. Figure 2 displays variations in vertical displacement for multiple points in all groups, comparing lesioned to control animals. The authors neglect to clarify which colors correspond to each group, making it impossible to interpret the results. Additionally, statistical significance indicators are provided for each bin, but an integrated analysis of the area under the curve for each group may be more appropriate. Furthermore, the percentage displayed at the bottom of each panel lacks an explanation of its significance.
4. The rationale for calculating the factor of dissimilarity in this dataset is inadequately explained and should be revisited.
5. There are instances of undefined acronyms that need clarification when first introduced in the text.
6. The document requires proofreading to enhance language clarity.
Addressing these issues will significantly improve the paper's overall quality and readability.
Comments on the Quality of English Language
The quality of English language must be improved
Author Response
Thanks for revising this paper; all comments and suggestions are considered, and changes were made. The manuscript was rewritten.
In the new version:
I.- the introduction was extended
II. We added one more reference about the treatment of traumatic brain injury
III. The introduction was restructured to clarify the paper's rationale
IV.- The methods were clarified, and the legend of Figure 1 Extended
Concerning comments:
1.- The presentation of data was reorganized for significant clarity.
a) We added a supplementary table to clarify the results.
2 .- Figure 1 was modified. We added a color symbol to clarify the graphs
3.- We made the statistical value for each bin of each step to develop further a control system that recovers the standard step displacement or electrically stimulates the adequate muscle to regain a standard step. An integrated value did not give us the precision of joint changes during the step cycle. The global changes could be estimated with the dissimilarity index.
4.- the rationale of the dissimilarity factor was clarified, and we added a supplementary figure
5. All acronyms are now well-defined.
6. English improved with the help of a native English speaker
Reviewer 2 Report
Comments and Suggestions for Authors
The paper by Alpirez et al. using a Model of Hippocampal Injury in rodents, provides important results. The manuscript is generally well structured. However, the quality of writing and the quality of some figures must be improved in order to enhance the quality of your paper.
Comments on the Quality of English Language
There are some spelling mistakes throughout the text. English proofreading is highly required.
Author Response
Thank you very much for reviewing this paper. All your comments were taken into account. The paper was rewritten to increase the quality of the paper.
In the new version:
I.- the introduction was extended
II. We added one more reference about treatment in traumatic brain injury
III. The introduction was restructured to clarify the paper's rationale
IV.- The methods were clarified, and the legend of Figure 1 extended
A native English speaker corrected the English language, and in the graph of figures, others were added to other illustrations to increase their clarity.
Reviewer 3 Report
Comments and Suggestions for Authors
The manuscript analyzes hindlimb's locomotion kinematics changes in response to hippocampal injury using the same methodology in rodents (mice and rats)
My comments
-The introduction needs to be divided into paragraphs
-The conclusin needs improvement. The involvement of sex and species changes should be highlighted
-Indicate the age of the animals used
-Figure 1 needs improvement. The images are hazzy and the changes on histological images (arrows, black arrows) should be verified in thr figure legend
Comments on the Quality of English Language
Minor editing of English language required
Author Response
Thanks very much for reviewing this paper.
- The introduction was extended to clarify the main points and paper rationale.
- The conclusion was modified and explained the importance of the results.
- The animal age was written in Material and Methods.
- Figure 1 was changed to clarify the main points
- The paper was rewritten with the help of a native English speaker
Round 2
Reviewer 1 Report
Comments and Suggestions for Authors
Authors have appropriately revised the manuscript.
Comments on the Quality of English Language
The overall presentation of the manuscript has been improved.
Reviewer 3 Report
Comments and Suggestions for Authors
The authors addressed my comments